# Coarse to Fine Segmentation Method Enables Accurate and Efficient Segmentation of Organs and Tumor in Abdominal CT

Hui Meng[1][0000−0003−4061−2100], Haochen Zhao[2][0009−0007−9234−3594], Deqian Yang[1][0009−0008−6633−3541], Songping Wang[2][0009−0001−4513−7284], and Zhenpeng Li[2][0009−0001−4214−3355]

[1] School of Intelligent Science and Technology, Hangzhou Institute for Advanced Study, University of Chinese Academy of Sciences, Hangzhou 310024, China
[2] Hangzhou Innovation Institute of Beihang University, Hangzhou 310051, China
huimeng@ucas.ac.cn

**Abstract.** Automatic segmentaion of organs and tumor in abdominal CT scans is essential for cancer diagnosis and treatment monitoring. However, there does not exist an accurate and efficient method for universal organ and tumor segmentation in abdominal CT scans. Therefore, we propose a coarse to fine segmentation (CFS) method based on pseudo labels. Specifically, the CFS consists of a coarse segmentation model (CSM), a tumor segmentation model (TSM), and an organ segmentation model (OSM). The CSM is trained to segment abdominal regions in CT scans. The TSM and the OSM are trained to generate segmentation masks of organs and tumor. The outputs of the TSM and the OSM are merged to generate the final segmentation results. To improve efficiency of the CFS, we optimize the inference process by streamlining intricate steps. On validation set of FLARE23 challenge, our method achieves mean DSC of 91.59% and mean NSD of 95.74% on organ segmentation, and mean DSC of 47.12% and mean NSD of 39.94% on tumor segmentation. The mean inference time is 24.12s, and the mean area under the GPU memory-time curve is 39543.46MB.

**Keywords:** segmentation · abdominal CT · coarse to fine.

## 1 Introduction

Abdominal organs are quite common cancer sites, such as colorectal cancer and pancreatic cancer, which are the 2nd and 3rd most common cause of cancer death [24]. Computed Tomography (CT) scanning is widely used for diagnosis and treatment monitoring of abdominal cancers. Nowadays, radiologists and clinicians measure the tumor and organs on CT scans based on manual two-dimensional measurements, which is inherently subjective with considerable inter- and intra-expert variability. Therefore, it is necessary to develop an automatic segmentation method for simultaneous segmentation of organs and tumor in abdominal CT scans.

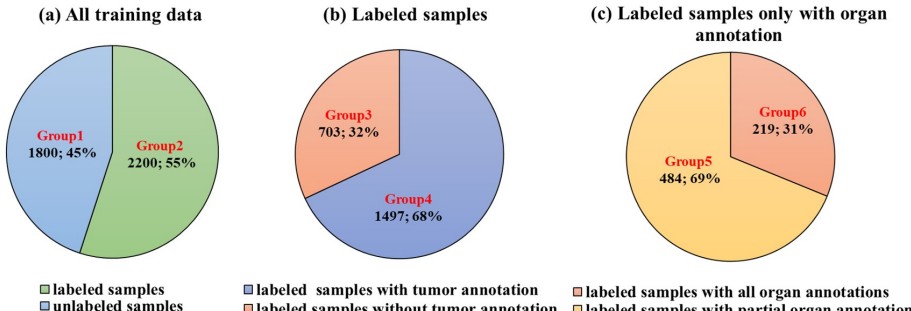

**Fig. 1.** Data distribution of training set in FLARE23 challenge. (a) Distribution of labeled samples and unlabeled samples. (b) Data partitioning of labeled samples. (c) Data partitioning of labeled samples only with organ annotation.

For segmentation of abdominal CT scans, the difficulty of data annotation leads to a lack of public large-scale labeled datasets. Thus, researchers try to improve segmentation accuracy using unlabeled data and partial-label data. To effectively utilize unlabeled data, Zhou et al. proposed a deep multiplane co-training approach [27] to generate dependable pseudo labels. Similarly, Lee et al. devised an advanced pseudo-label quality discriminator to effectively guide and regulate network learning of unlabeled data [17]. Additionally, participants of FLARE22 challenge proposed many semi-supervised methods for organ segmentation in abdominal CT scans, including pseudo label-based methods [13,1,6], consistency learning-based methods [9,16,22], and cross pseudo supervision-based methods [4,12]. Compared with the consistency learning-based methods and the cross pseudo supervision-based methods, the pseudo label-based methods achieved more accurate segmentation results.

Although the above researchs improved semi-supervised segmentation accuracy, the tasks of them mainly focus on segmentation of organs or one type of tumor whose distribution in CT scans is relatively fixed. In FLARE23 challenge, the segmentation task focuses on 13 organ segmentation and pan-cancer segmentation. Different from existing researchs, the FLARE23 aims to segment various abdominal cancer types and 13 organs simultaneously. Compared with organ segmentation, the pan-cancer segmentation faces two major challenges. The first one is that the tumor location, tumor shape, and the tumor number are various among different CT scans, which make the tumor feature complex. The second one is lack of cases with both organ and tumor annotations. Therefore, it is necessary to make full use of cases with tumor annotation.

To achieve accurate organ and pan-cancer segmentation, we propose a coarse to fine segmentation (CFS) method based on pseudo labels. The CFS consists of a coarse segmentation model (CSM), a tumor segmentation model (TSM), and an organ segmentation model (OSM), whose architectures are all nnU-Net [15]. The CSM is first trained using cases with full organ annotations. Then, cases with only tumor labels are inferenced to generate pseudo labels of organs based on the trained CSM. Next, we use the cases with tumor labels and pseudo labels of organs to train the TSM. The OSM is trained using all 4000 cases with

ground truth and pseudo labels generated by the [14]. During inference, we first segment input CT scans using the CSM to obtain abdominal regions. Then, the abdominal regions are segmented by the TSM and the OSM separately. Last, the segmentation masks given by the TSM and the OSM are merged to generate the final segmentation results. To improve efficiency of the CFS, we optimize the inference process by streamlining intricate steps.

## 2   Method

### 2.1   Preprocessing

In our method, preprocessing operations include data grouping, image cropping, data resampling, intensity normalization, and data augmentation. The details of the preprocessing operations are listed as follows:

– Data grouping:
  The training set is analysed and grouped based on annotations. As shown in Fig. 1(a), the training set consists of 1800 unlabeled samples ($Group$ 1) and 2200 labeled samples ($Group$ 2). For the $Group$ 2, we further divide labeled samples based on whether annotations contain tumor labels. As shown in Fig. 1(b), there are 703 labeled samples without tumor annotations ($Group$ 3) and 1497 labeled samples with tumor annotations ($Group$ 4). Furthermore, we divide the $Group$ 3 into 484 samples with partial organ annotations ($Group$ 5) and 219 samples with all organ annotations ($Group$ 6) (Fig. 1(c)).
– Cropping strategy:
  Before model training, the training CT scans are cropped along the z-axis direction based on ground truth or pseudo labels. Specifically, the indices of start slice and the end slice of region containing targets are first calculated based on labels. To reserve context information of segmentation targets, we reduce the index of the start slice by 10 and add the index of the end slice by 10. During model training, the cropped CT scans are further cropped based on non-zero region introduced by nnU-Net [15].
– Resampling method for anisotropic data:
  We perform image redirection to the desired orientation, followed by resampling all CT scans to match the median voxel spacing of the training dataset. Specifically, third-order spline interpolation is used for image resampling, and the nearest neighbor interpolation is employed for label resampling.
– Intensity normalization approach:
  We gather pixel values in the cropped CT scans and subsequently truncate all data to fall within [0.5, 99.5] of foreground voxel values. Following that, z-score normalization is applied.
– Data augmentation:
  In our method, random rotation, random scaling, elastic transformation, brightness transformation, contrast transformation, and gamma transformation are used for data augmentation.

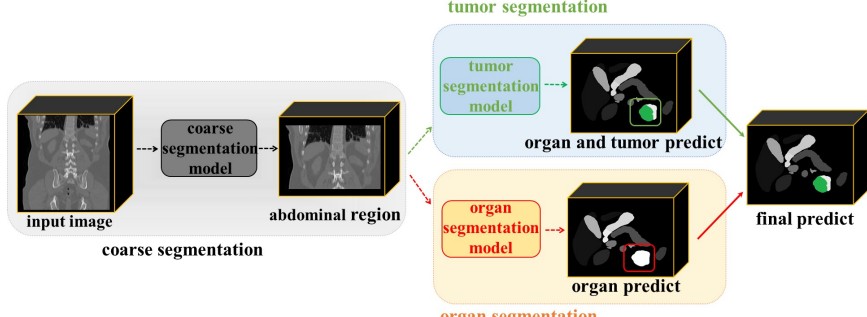

**Fig. 2.** The overall architecture of the CFS method. The CFS consists of three models: the CSM, the TSM, and the OSM. The three models are all based on nnU-Net architecture, but trained with different data.

## 2.2   Proposed Method

As shown in Fig. 2, the architecture of our proposed method contains three segmentation models, i.e. the CSM, the TSM, and the OSM. Noted that the three segmentation models are all based on the nnU-Net framework [15]. We combine Dice loss and cross-entropy loss to train all models. The details of model training and inference process are described in the following.

**Training of the CSM** To reduce redundant information in CT scans and extract abdominal regions, we first train the CSM using 219 labeled samples with all organ annotations (*Group* 6). Based on the trained CSM, we predict pseudo labels of 13 organs on *Group* $1 - 5$ and merge them with the corresponding ground truth. Then, we obtain 4000 training data with all organ annotations (real labels or pseudo labels), which is termed as *Dataset* 1.

**Training of the TSM** To train the TSM, we first extract the 1497 samples of the *Group* 4 from *Dataset* 1. The 1497 samples are randomly divided into training data and validation data by 4:1. The TSM is trained to segment 13 organs and tumor. The best weights are saved every 50 epochs based on the highest segmentation accuracy of tumor on validation data. Then, we evaluate the saved weights on online validation set (100 samples) and select the best weight (*tumor_weight*1) with the highest segmentation accuracy of tumor. To further improve accuracy of tumor segmentation, we train the TSM with random initialization again and save another weight (*tumor_weight*2) following the above method.

**Training of the OSM** The OSM is trained using all training data with the pseudo labels generated by the FLARE22 winning algorithm [14]. The 4000 samples are randomly divided into training set and validation set by 4:1. The model is trained to segment 13 organs and the best weights are saved based on the performance of validation set every 50 epochs. Finally, we evaluate the saved weights on the online validation and choose the weight with the highest segmentation accuracy as the weight of the OSM.

**Inference process** During inference, the test CT scans are first segmented by the CSM, and the abdominal regions of the CT scans are cropped based on the

segmentation masks of 13 organs. Then, the abdominal regions are segmented by the TSM with $tumor\_weight1$ and $tumor\_weight2$, respectively. The segmentation masks given by the two weights are merged to obtain the masks of tumor and 13 organs ($mask1$). Next, the abdominal regions are segmented by the OSM and the masks of 13 organs are generated ($mask2$). Finally, we modify the $mask1$ with $mask2$ for organ labels and obtain the final segmentation results.

To improve the efficiency of the CFS, we optimize the inference process of conventional nnU-Net by streamlining intricate steps while ensuring faster sampling without compromising accuracy. During the employment of sliding window inference, we omit Gaussian weighting and mirror inference. Compared with the conventional inference of the nnU-Net, our inference strategy achieves a significant reduction in inference time. To reduce resource consumption, we use abdominal regions rather than whole CT scans as input of the TSM and the OSM.

### 2.3   Post-processing

Based on the segmentation mask, we retain the largest connected region for each segmentation organ based on centroid distances. The small connected regions of segmented organs are removed to reduce false positive islands. Noted that the segmented tumor regions are not processed because there are might multiple tumors in one CT scan.

## 3   Experiments

### 3.1   Dataset and evaluation measures

The FLARE 2023 challenge is an extension of the FLARE 2021-2022 [20][21], aiming to promote the development of foundation models in abdominal disease analysis. The segmentation targets cover 13 organs and various abdominal lesions. The training dataset is curated from more than 30 medical centers under the license permission, including TCIA [5], LiTS [3], MSD [2], KiTS [10,11], autoPET [8,7], TotalSegmentator [25], and AbdomenCT-1K [19]. The training set includes 4000 abdomen CT scans where 2200 CT scans with partial labels and 1800 CT scans without labels. The validation and testing sets include 100 and 400 CT scans, respectively, which cover various abdominal cancer types, such as liver cancer, kidney cancer, pancreas cancer, colon cancer, gastric cancer, and so on. The organ annotation process used ITK-SNAP [26], nnU-Net [15], and MedSAM [18].

The evaluation metrics encompass two accuracy measures—Dice Similarity Coefficient (DSC) and Normalized Surface Dice (NSD)—alongside two efficiency measures—running time and area under the GPU memory-time curve. These metrics collectively contribute to the ranking computation. Furthermore, the running time and GPU memory consumption are considered within tolerances of 15 seconds and 4 GB, respectively.

### 3.2   Implementation details

**Environment settings**  The development environments and requirements are presented in Table 1.

**Table 1.** Development environments and requirements.

| | |
|---|---|
| System | Ubuntu 20.04 |
| CPU | 13th Gen Intel(R) Core(TM) i7-13700KF 3.40 GHz |
| RAM | 8×4GB; 2.67MT/s |
| GPU (number and type) | NVIDIA 4070Ti 12G |
| CUDA version | 12.1 |
| Programming language | Python 3.10 |
| Deep learning framework | Pytorch (Torch 1.12, torchvision 0.2.3) |
| Specific dependencies | nnU-Net 1.7.0 |
| Code | https://github.com/zhuji423/MICCAI2023_Flare2023 |

**Table 2.** Training protocols for the three segmentation models.

| | CSM | TSM | OSM |
|---|---|---|---|
| Network initialization | "he" normal initialization | | |
| Batch size | 2 | 2 | 2 |
| Patch size | 64×160×224 | 96×128×192 | 96×160×192 |
| Total epochs | 1500 | 2500 | 2000 |
| Optimizer | SGD with nesterov momentum ($\mu = 0.99$) | | |
| Initial learning rate (lr) | 0.01 | 0.01 | 0.01 |
| Lr decay schedule | halved by 200 epochs | | |
| Training time | 48 hours | 65 hours | 55.5 hours |
| Number of model parameters | 41.22M | 41.22M | 41.22M |
| Number of flops | 59.32G | 59.32G | 59.32G |
| $CO_2$eq | 1.23 Kg | 2.34 Kg | 1.56 Kg |

**Training protocols**  The data processing of unlabeled data and samples with partial labels has been described in Section 2.2. We adopt nnU-Net's data augmentation to train the CSM, the TSM, and the OSM. The best weights of the three models are determined based on the online validation, and the details have been introduced in Section 2.2. The detailed training protocols of the three models are listed in Table 2.

## 4   Results and discussion

In this section, we assess the CFS method using FLARE23 dataset. Ablation studies on utilization of unlabeled data and effectiveness of the three models are conducted. We present quantitative results, qualitative results, and efficiency results on validation set. Additionally, results on final testing set are presented. Last, we discuss limitations of the CFS method and our future work.

**Table 3.** Quantitative evaluation results of CFS method.

| Target | Public Validation | | Online Validation | | Testing | |
|---|---|---|---|---|---|---|
| | DSC(%) | NSD(%) | DSC(%) | NSD(%) | DSC(%) | NSD (%) |
| Liver | 98.62 ± 0.40 | 99.28 ± 0.89 | 98.56 | 99.23 | 97.83 | 98.17 |
| Right Kidney | 96.59 ± 5.46 | 96.54 ± 0.41 | 95.65 | 95.84 | 95.74 | 95.52 |
| Spleen | 97.98 ± 4.09 | 98.69 ± 4.04 | 97.07 | 97.96 | 97.76 | 98.51 |
| Pancreas | 87.88 ± 4.88 | 97.08 ± 3.65 | 86.71 | 96.19 | 91.32 | 97.22 |
| Aorta | 97.62 ± 1.15 | 99.42 ± 0.40 | 97.57 | 99.31 | 98.12 | 99.73 |
| Inferior vena cava | 92.33 ± 8.52 | 93.17 ± 8.81 | 92.13 | 92.74 | 92.70 | 94.12 |
| Right adrenal gland | 87.03 ± 12.47 | 95.91 ± 13.32 | 87.40 | 96.52 | 88.26 | 96.73 |
| Left adrenal gland | 88.82 ± 5.31 | 97.70 ± 3.09 | 88.06 | 96.7 | 89.32 | 96.77 |
| Gallbladder | 88.49 ± 19.27 | 89.68 ± 20.6 | 89.01 | 89.92 | 86.14 | 87.90 |
| Esophagus | 83.03 ± 16.47 | 91.38 ± 16.40 | 83.59 | 92.36 | 89.50 | 87.90 |
| Stomach | 94.37 ± 4.72 | 97.07 ± 4.83 | 94.83 | 97.34 | 97.46 | 95.64 |
| Duodenum | 84.57 ± 8.48 | 94.96 ± 6.45 | 84.80 | 94.99 | 96.83 | 89.36 |
| Left kidney | 95.03 ± 11.56 | 94.70 ± 13.59 | 95.26 | 95.54 | 95.16 | 94.75 |
| Tumor | 54.45 ± 34.80 | 47.31 ±31.78 | 47.12 | 39.94 | 63.43 | 51.02 |
| Average | 89.05 ± 9.82 | 92.34 ± 9.72 | 91.59 | 95.74 | 92.64 | 96.10 |

## 4.1 Quantitative results on validation set

In our method, the CSM is trained using 219 labeled data with all organ annotations. The TSM is trained using 1497 labeled data with tumor annotation and pseudo labels of 13 organs. To further improve the segmentation performance, we train the OSM using all 4000 CT scans including unlabeled data. To verify the effectiveness of the utilization of the unlabeled data, we evaluate the segmentation results given by the CSM and the TSM (two-stage method). The mean and standard deviation (SD) of DSC and NSD on 50 validation data (public validation) are calculated based on the official evalution code. The mean of DSC and NSD on 100 validation data are calculated on CodaLab platform.

**Table 4.** Quantitative evaluation results of two-stage method.

| Target | Public Validation | | Online Validation | |
|---|---|---|---|---|
| | DSC(%) | NSD(%) | DSC(%) | NSD(%) |
| Liver | 95.49 ± 5.94 | 99.09 ± 0.89 | 97.21 | 98.59 |
| Right Kidney | 96.59 ± 5.46 | 96.54 ± 0.41 | 93.74 | 93.57 |
| Spleen | 96.22 ± 10.12 | 97.37 ± 10.44 | 94.28 | 96.26 |
| Pancreas | 86.63 ± 6.96 | 96.38 ± 6.16 | 84.45 | 95.54 |
| Aorta | 96.78 ± 2.43 | 98.6 ± 3.54 | 95.98 | 97.88 |
| Inferior vena cava | 92.84 ± 6.77 | 94.41 ± 7.20 | 93.00 | 95.28 |
| Right adrenal gland | 84.10 ± 15.2 | 94.25 ± 16.42 | 81.73 | 93.67 |
| Left adrenal gland | 85.01 ± 10.54 | 95.72 ± 10.04 | 80.63 | 93.17 |
| Gallbladder | 85.44 ± 22.4 | 86.52 ± 23.53 | 79.96 | 80.19 |
| Esophagus | 82.13 ± 16.26 | 91.13 ± 16.23 | 81.98 | 92.54 |
| Stomach | 93.83 ± 4.68 | 97.16 ± 4.68 | 93.54 | 97.53 |
| Duodenum | 83.74 ± 7.81 | 94.84 ± 5.76 | 82.85 | 94.67 |
| Left kidney | 94.12 ± 10.23 | 93.67 ± 12.91 | 92.91 | 92.89 |
| Tumor | 54.45 ± 34.8 | 47.31 ± 31.78 | 47.12 | 39.94 |
| Average | 87.66 ± 11.4 | 91.64 ± 9.72 | 85.67 | 90.12 |

Table. 3 and Table. 4 list quantitative resutls of the CFS method and the two-stage method, respectively. Because pseudo labels of unlabeled data lack tumor label, the tumor segmentation accuracy of the CFS method is same as that of the two-stage method. Compared with the two-stage method, the organ segmentation accuracy is improved by the CFS method. For right adrenal gland, left adrenal gland, and gallbladder, the CFS method yields results of 87.4%, 88.06%, and 89.01% in mean DSC on online validation, which outperforms the two-stage method by 5.67%, 7.43%, and 9.05%, respectively. These results demonstrate that the unlabeled data have great potential to improve segmentation accuracy of organs, especially small organs.

Furthermore, we implement an ablation study to evaluate the effectiveness of the three models in our method. The quantitative results are list in Table. 5. Compared with the CSM, the TSM significantly improves the mean organ DSC and the mean organ NSD. Additionally, the OSM achieves higher organ DSC and higher organ NSD than the two-stage method. Furthermore, we evaluate the effectiveness of the two weights used in the TSM. The quantitative results (Table. 6) demonstrate that inference using two weights obtains the highest tumor segmentation accuracy. All these results illustrate the effectiveness of the TSM and the OSM.

**Table 5.** Quantitative results of ablation study on the three models.

| CSM | TSM | OSM | organ_DSC | organ_NSD | tumor_DSC | tumor_NSD |
|-----|-----|-----|-----------|-----------|-----------|-----------|
| √ | | | 66.97% | 70.19% | \ | \ |
| √ | √ | | 90.35% | 95.70% | 47.12% | 39.94% |
| √ | √ | √ | 91.59% | 95.74% | 47.12% | 39.94% |

**Table 6.** Quantitative results of ablation study on tumor segmentation weights.

| tumor_weight1 | tumor_weight2 | organ_DSC | organ_NSD | tumor_DSC | tumor_NSD |
|---------------|---------------|-----------|-----------|-----------|-----------|
| √ | | 91.58% | 95.75% | 43.09% | 36.87% |
| | √ | 91.60% | 95.75% | 46.14% | 38.82% |
| √ | √ | 91.59% | 95.74% | 47.12% | 39.94% |

### 4.2   Qualitative results on validation set

Fig. 3 shows two examples with good segmentation results and two examples with bad segmentation results given by the CFS and the two-stage method, respectively. In Case #FLARETs_0019 (slice #155) and Case #FLARETs_0099 (slice #290), both the two-stage method and the CFS achieve accurate organ segmentation. However, the two-stage method and the CFS fail to segment small organs accurately in Case #FLARETs_0001 (slice #99) and Case #FLARETs_0029 (slice #290). Additionally, we present three examples with bad organ segmentation results in Fig. 4. In Case #FLARETs_0001 (slice #31), stomach (green label) and pancreas (yellow label) are not completely segmented. In Case #FLARETs_0029 (slice #290), duodenum in the upper right corner and gallbladder are not segmented. Additionally, the stomach and the pancreas are not fully segmented in the Case #FLARETs_0011 (slice #97). These results indicate that our method still has room for improvement in organ segmentation, especially in small organ segmentation.

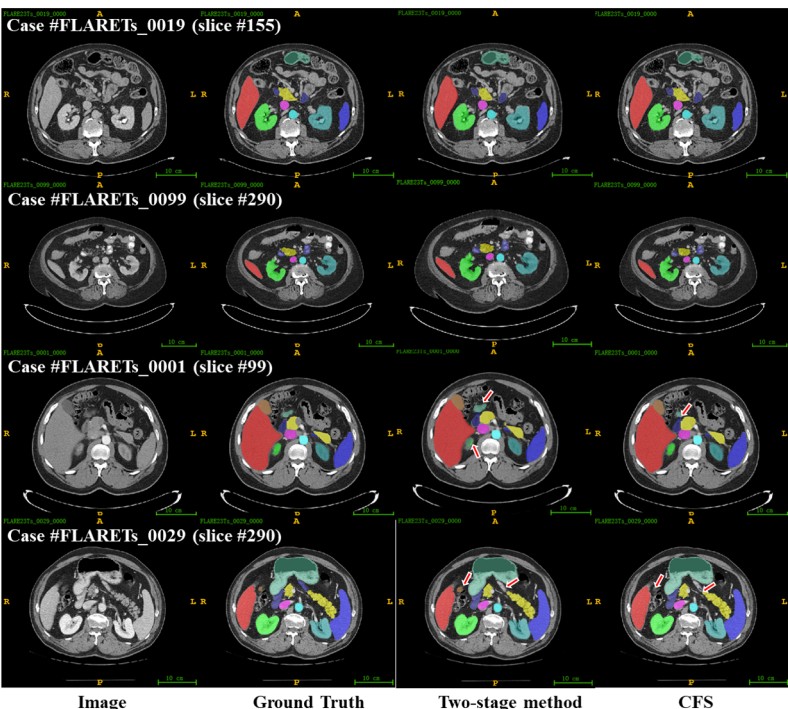

**Fig. 3.** Qualitative results given by the two-stage method and the CFS, respectively. Case #FLARETs_0019 and Case #FLARETs_0099 are examples with good segmentation results. Case #FLARETs_0001 and Case #FLARETs_0029 are examples with bad segmentation results. Red arrows indicate regions with bad segmentation.

Besides qualitative results of organ segmentation, we present three examples with tumor segmentation results in Fig. 5. In Case #FLARETs_0051 (slice #73), the tumor in the right kidney is segmented as liver by the CFS. In Case #FLARETs_0071 (slice #104), the tumor in the liver is not segmented. Additionally, the tumor in the Case #FLARETs_0048 (slice #297) is segmented as liver or stomach. These results demonstrate that tumors have similar features with organs and it is difficult for the CFS to segment tumors accurately.

### 4.3   Segmentation efficiency results on validation set

Table. 7 lists quantitative efficiency results in terms of the running time and GPU memory consumption. Total GPU denotes the area under GPU Memory-Time curve. Evaluation GPU platform is NVIDIA QUADRO RTX5000 (16G). The average running time of the CFS is 24.12s, while the two-stage method obtains shorter average running time (18.17s). Additionally, the GPU memory consumption of the two-stage method and the CFS is both within 4GB. Furthermore, quantitative efficiency results of eight examples given by the CFS and the two-stage method are listed in Table. 8 and Table. 9, respectively. All results demonstrate that our methods achieve efficient segmentation.

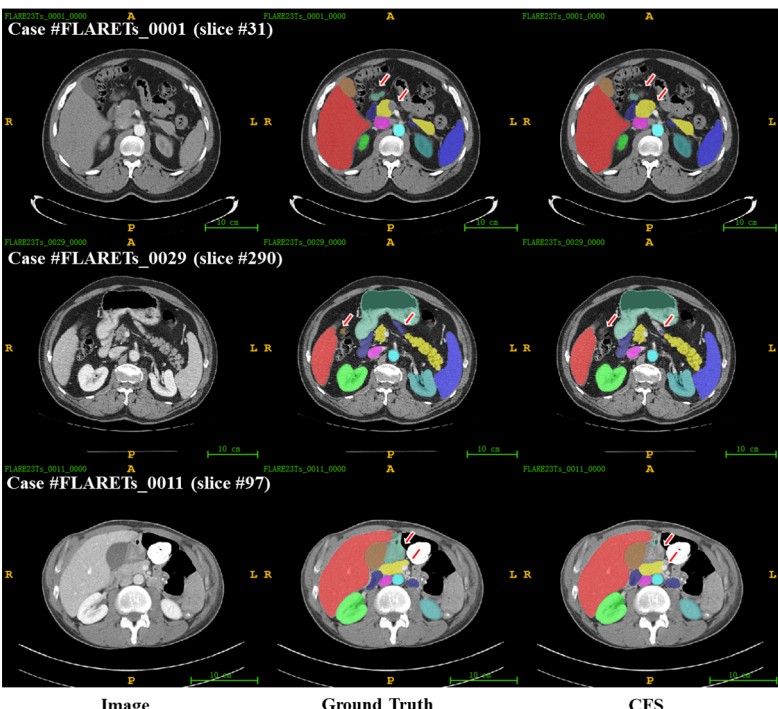

**Fig. 4.** Qualitative results of three examples with bad organ segmentation results given by the CFS. Red arrows indicate regions with bad segmentation.

**Table 7.** Quantitative efficiency results on online validation (Mean ± SD).

| Efficiency index | two-stage method | CFS |
|---|---|---|
| Running Time (s) | 18.17 ± 4.78 | 24.12 ± 5.39 |
| Max GPU (MB) | 3137.16 ± 232.18 | 3480.48 ± 150.48 |
| Total GPU (MB) | 26314.9 ± 8457.26 | 39543.46 ± 9845.58 |

**Table 8.** Quantitative efficiency results given by the CFS.

| Case ID | Image Size | Running Time (s) | Max GPU (MB) | Total GPU (MB) |
|---|---|---|---|---|
| 0001 | (512, 512, 55) | 21 | 3452 | 36129 |
| 0051 | (512, 512, 100) | 26.31 | 3452 | 47694 |
| 0017 | (512, 512, 150) | 26.24 | 3452 | 46555 |
| 0019 | (512, 512, 215) | 24.03 | 3452 | 37880 |
| 0099 | (512, 512, 334) | 28.09 | 3452 | 42890 |
| 0063 | (512, 512, 448) | 34.86 | 3452 | 54187 |
| 0048 | (512, 512, 499) | 35.65 | 3452 | 52391 |
| 0029 | (512, 512, 554) | 44.23 | 4064 | 69970 |

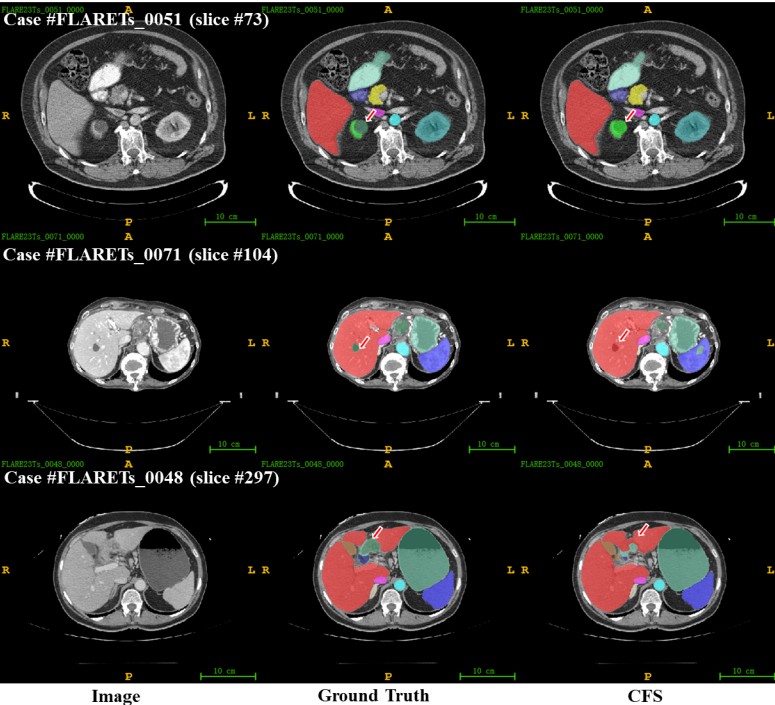

**Fig. 5.** Qualitative results of three examples with tumor segmentation results given by the CFS. Red arrows indicate tumor regions.

**Table 9.** Quantitative efficiency results given by the two-stage method.

| Case ID | Image Size | Running Time (s) | Max GPU (MB) | Total GPU (MB) |
|---|---|---|---|---|
| 0001 | (512, 512, 55) | 18.26 | 3071 | 23535 |
| 0051 | (512, 512, 100) | 23.67 | 3071 | 39243 |
| 0017 | (512, 512, 150) | 22.97 | 3097 | 37210 |
| 0019 | (512, 512, 215) | 17.45 | 3071 | 23727 |
| 0099 | (512, 512, 334) | 22.51 | 3071 | 33945 |
| 0063 | (512, 512, 448) | 26.1 | 3229 | 37647 |
| 0048 | (512, 512, 499) | 23.59 | 3223 | 31547 |
| 0029 | (512, 512, 554) | 34.15 | 4025 | 49582 |

### 4.4   Results on final testing set

The quantitative results of the CFS on final testing set are listed in Table. 3. The mean DSC and the mean NSD are 92.64% and 96.10%, respectively. Compared with the results on online validation set, the CFS achieves higher tumor DSC (63.43%) and higher tumor NSD (51.02%) on the final testing set. Additionally, the CFS obtains mean organ DSC of 92.64% and mean organ NSD of 96.10% on the final testing set, respectively.

### 4.5   Limitation and future work

The major limitation of the CFS is that the tumor segmentation accuracy is much lower than the organ segmentation accuracy. It is valuable to propose novel methods to improve the tumor segmentation accuracy. Besides that, the training of the CFS is cumbersome because the three models are trained seperately. Furthermore, the segmentation accuracy of the small organs is low, and the robustness of the segmentation results is poor.

In the future, we will continue working on segmentation of abdominal organs and tumor in CT scans. We will further investigate semi-supervised methods for segmentation of abdominal CT scans. Specifically, we will mainly focus on improvement of tumor segmentation and small organ segmentation.

## 5   Conclusion

In this study, we propose a novel CFS method for multi-organ and tumor segmentation in abdominal CT scans. The CFS consists of the CSM, the TSM, and the OSM, which are trained with different data. During inference, test samples are first segmented by the CSM to obtain abdominal regions. Then, the abdominal regions are segmented by the TSM and the OSM, respectively. Finally, the segmentation masks of the TSM and the OSM are merged to generate the final segmentation results. Besides that, we optimize the inference process by streamlining intricate steps to improve the efficiency of the CFS.

To validate segmentation performance of the CFS, we implement ablation studies on utilization of unlabeled data and effectiveness of the three models. The experimental results demonstrate that the unlabeled data can improve segmentation accuracy of organs, especially small organs. Additionally, the TSM achieves higher tumor segmentation accuracy using two weights than using one weight. The organ segmentaion model further improves the organ segmentation accuracy given by the TSM. Furthermore, the quantitative results of segmentation efficiency demonstrate that the two-stage method and the CFS achieve fast multi-organ and tumor segmentation in CT scans.

**Acknowledgements**  The authors of this paper declare that the segmentation method they implemented for participation in the FLARE 2023 challenge has not used any pre-trained models nor additional datasets other than those provided by the organizers. The proposed solution is fully automatic without any manual intervention. We thank all the data owners for making the CT scans publicly available and CodaLab [23] for hosting the challenge platform. This work was supported in part by the National Natural Science Foundation of China under Grant Nos. 62303127 and 62273009.

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
