# OpenReview forum: "Coarse to Fine Segmentation Method Enables Accurate and Efficient Segmentation of Organs and Tumor in Abdominal CT"
_MICCAI.org/2023/FLARE — Submitted to FLARE 2023_

### Official Review · Reviewer_dAMh · 2023-10-20
**Enhancing Tumor and Organ Segmentation through Coarse-to-Fine Segmentation with Pseudo Labels and Process Optimization (CFS)**

**Rating:** 6
**Confidence:** 4

**Review:**

Pros:
1.The proposed method of coarse-to-fine segmentation (CFS), based on pseudo labels and optimization of the inference process by streamlining intricate steps, has achieved good results in tumor and organ segmentation.

Cons：
1.please check the Fig4, there are blank area in the pictures.

---

### Official Review · Reviewer_YVUL · 2023-10-25
**Coarse to Fine Segmentation Method Enables Accurate and Efficient Segmentation of Organs and Tumor in Abdominal CT**

**Rating:** 10
**Confidence:** 5

**Review:**

Why the numbers of model parameters and flops are blank for CSM and OSM in Table 2

---

### Official Review · Reviewer_s4Rz · 2023-10-25

**Rating:** 7
**Confidence:** 5

**Review:**

In Figure 2, the font should be Times New Roman.

Loss function is missing in the Method section.

---

### Decision · Program_Chairs · 2023-10-25

Accept